# Dietary Characteristics and Influencing Factors on Chinese Immigrants in Canada and the United States: A Scoping Review

**DOI:** 10.3390/nu14102166

**Published:** 2022-05-23

**Authors:** Ping Zou, Dong Ba, Yan Luo, Yeqin Yang, Chunmei Zhang, Hui Zhang, Yao Wang

**Affiliations:** 1School of Nursing, Nipissing University, 222 St. Patrick Street, Suite 618, Toronto, ON M5T 1V4, Canada; 2Faculty of Health Sciences, McMaster University, 1280 Main Street West, Hamilton, ON L8S 4L8, Canada; bad@mcmaster.ca; 3Faculty of Nursing, Health Science Center, Xi’an Jiaotong University, No. 76 Yanta West Road, Xi’an 710061, China; luoyan0904@xjtu.edu.cn; 4School of Nursing, Wenzhou Medical University, Chashan Higher Education Park, Wenzhou 325035, China; yangyq@wmu.edu.cn (Y.Y.); sallyzcm@wmu.edu.cn (C.Z.); 5Department of Cardiology, Guizhou Provincial People’s Hospital, Guiyang 550002, China; zhanghui88640@163.com; 6Xiang Ya School of Nursing, Central South University, Changsha 410013, China

**Keywords:** diet, influencing factors, Chinese, immigrants, review, Canada, United States

## Abstract

Background: Chinese immigrants are an integral part of Canadian and American society. Chinese immigrants believe diet to be an important aspect of health, and dietary behaviours in this population have been associated with changes in disease risk factors and disease incidence. This review aims to summarize the characteristics of the dietary behaviours of Chinese immigrants and the associated influencing factors to better inform individual, clinical, and policy decisions. Methods: This scoping review was written in accordance with PRISMA guidelines. MEDLINE, PsychINFO, CINAHL, AgeLine, ERIC, ProQuest, Nursing and Allied Health Database, PsychARTICLES, and Sociology Database were utilized for the literature search. Articles were included if they explored dietary or nutritional intake or its influencing factors for Chinese immigrants to Canada or the United States. Results: A total of 51 papers were included in this review. Among Chinese immigrants in Canada and the United States, the intake of fruits and vegetables, milk and alternatives, and fiber were inadequate against national recommendations. Chinese immigrants showed increased total consumption of food across all food groups and adoption of Western food items. Total caloric intake, meat and alternatives intake, and carbohydrate intake increased with acculturation. Individual factors (demographics, individual preferences, and nutritional awareness), familial factors (familial preferences and values, having young children in the family, and household food environment), and community factors (accessibility and cultural conceptualizations of health and eating) influenced dietary behaviours of Chinese immigrants. Discussion and Conclusion: Efforts should be undertaken to increase fruit, vegetable, and fibre consumption in this population. As dietary acculturation is inevitable, efforts must also be undertaken to ensure that healthy Western foods are adopted. It is important for healthcare providers to remain culturally sensitive when providing dietary recommendations. This can be achieved through encouragement of healthy ethnocultural foods and acknowledgement and incorporation of traditional health beliefs and values into Western evidence-based principles where possible.

## 1. Introduction

Chinese immigrants are an integral part of the functioning North American society. According to the 2016 Canadian census, Chinese-born individuals represented the second largest visible minority group in Canada, with a population exceeding 1.5 million [1]. In the United States, there are nearly 2.5 million Chinese immigrants, with China being the top country of origin for immigrants [2]. Chinese immigrants to North America have been demonstrated to hold significant human capital [3] and have been regarded as a key minority group in society. Chinese immigrants usually maintain their cultural beliefs and values after immigration [4].

Chinese immigrants identify diet as an important factor in health maintenance and disease mitigation and management [5]. Traditional Chinese Medicine emphasizes the nutritional use of foods, herbs, and the concept of humoral balance between hot and cold foods to achieve good, stable health [6,7]. Additionally, the promotion of healthy dietary behaviours in the Chinese immigrant population have been shown to influence cardiovascular risk factors, including HDL cholesterol, blood pressure, and waist circumference, as well as cardiovascular disease incidence [8]. The impact of dietary factors on health outcomes has been well documented in the existing literature, with dietary and lifestyle behaviours being key to managing a variety of chronic diseases [9].

There is a need to summarize and identify the influencing factors of Chinese immigrants’ dietary behaviours to facilitate the design and implementation of specific dietary interventions in local North American communities. An existing scoping review discussed the nutritional health of Canadian immigrants of diverse ethnicities. They identified a dearth of evidence regarding how acculturation influences dietary habits [10]. This review included but was not specific to Chinese immigrants and missed psychosocial factors other than acculturation. A more focused narrative review on dietary habits of South Asians in Western countries identifies the examination of the intake of food groups as well as the intake of macro- and micro-nutrients as a key future step [11]. Moreover, a literature review of dietary patterns and influencing for Arabic-speaking immigrants and refugees in Western societies highlighted a gap in nutrition research for their respective immigrant groups [12]. Two literature reviews, which focused on Chinese immigrant populations, examined dietary behaviours in the context of specific diseases such as diabetes and focused on cultural influence [13,14]. Thus, there is a need to better synthesize the evidence on the specific dietary practices, changes, and influencing factors of Chinese immigrant groups.

This review strives to summarize the characteristics and influencing factors of dietary behaviours in the Chinese immigrant populations in Canada and the United States. The ecosocial theory proposes a framework of intertwined factors from micro to macro levels [15]. This framework has been previously applied in qualitative studies assessing influencing factors on dietary behaviours in Chinese immigrants with hypertension [16]. This present review employs the theory to elucidate the influencing factors on dietary behaviours among Chinese immigrants at personal, familial, and community levels. The research questions of this review are: (1) what are characteristics of diet behaviours among Chinese immigrants in Canada and the United States; (2) what are the factors influencing their dietary behaviours and changes?

## 2. Methods

The protocol and reporting of the results of this scoping review were based on the PRISMA statement [17].

### 2.1. Eligibility Criteria

Studies were included if they measured dietary or nutritional intake of Chinese immigrant adults to Canada or the United States, or if they investigated influencing factors of dietary behaviours for Chinese immigrant adults to Canada or the United States. We included qualitative, quantitative, and mixed-method studies.

Studies were excluded if they: (a) had results not specific to the population of Chinese adult immigrants to Canada and the United States, (b) did not have data relevant to dietary influencing factors or dietary characteristics, (c) were theses or review papers, or (e) did not have an accessible electronic text document.

### 2.2. Information Sources

Various health-related, psychological, sociological, and educational science databases, including MEDLINE, PsycINFO, CINAHL, AgeLine, ERIC, ProQuest, Nursing and Allied Health Database, PsycARTICLES, Sociology Database, and Education Research Complete, were selected for the literature search.

### 2.3. Search Strategy and Selection of Evidence

The databases were systematically searched using a combination of the following keywords: (Chin*) AND (immigr* OR migrant* or migrat*) AND (North America* OR Canad* OR United States OR America* OR New York* OR Toronto* OR California* OR Vancouver* OR Montreal) AND (Diet* OR food* OR eat* OR cook* OR nutri* OR carbohydrate* OR Protein* OR Fat or Fats OR Milk* OR Dairy OR Sugar* OR Potassium* OR Calcium* OR Sodium* OR Vitamin* Or fruit* Or vegetable* OR meat* OR rice* OR grain*)) OR ti((Chin*) AND (immigr* OR migrant* or migrat*) AND (North America* OR Canad* OR United States OR America* OR New York* OR Toronto* OR California* OR Vancouver* OR Montreal) AND (Diet* OR food* OR eat* OR cook* OR nutri* OR carbohydrate* OR Protein* OR Fat or Fats OR Milk* OR Dairy OR Sugar* OR Potassium* OR Calcium* OR Sodium* OR Vitamin* OR fruit* OR vegetable* OR meat* OR rice* OR grain*). The citations were exported into EndNote to remove any duplicates. The titles and abstracts of all citations were screened for relevance based on the established eligibility criteria. All eligible articles were searched for full text documents and the full text documents were carefully reviewed, with reasons for exclusion noted. Furthermore, tables of contents of key journals were hand-searched for the previous 2 years, and the reference lists of all eligible articles were manually searched. The most recent search was conducted in February 2022.

### 2.4. Quality Assessment

The Critical Appraisal Skills Program Checklists have been used as quality assessment tools for the included articles [18]. These checklists are not designed to generate a final quantitative score. Rather, they draw attention to the elements of a rigorous study and evaluate the study as a whole. Using these checklists, we were able to classify the quality of included papers as low, moderate, or high. Two researchers (DB & PZ) independently evaluated each article; any discrepancies in ratings were discussed along with the guidelines until consensus was reached. Papers rated as low quality were excluded. Thus, all papers included in this review were moderate to high quality.

### 2.5. Data Extraction

Data were independently extracted by two reviewers (DB & PZ) based on pre-determined criteria. From each article, various data, including authors, year of publication, study population, research design, recruitment method, sample size, sample characteristics, comparison group, features of the intervention, outcomes, measurements, significant findings, limitations, and future direction were extracted. The data were collected and organized into an Excel spreadsheet. The reviewers discussed disagreements in data extraction until consensus was reached.

### 2.6. Synthesis of Results

Once the data was organized in Excel, descriptive statistics were used to present the characteristics of the included studies. Thematic analysis was then used to summarize the findings of each research question. Categorization results were compared among reviewers (DB & PZ) and any disagreements among reviewers were resolved with a consensus decision. Due to the heterogeneity of the measurement tools used by the included studies, a meta-analysis was not performed since attempting to combine different measurements for the same variable would be inappropriate.

## 3. Results

### 3.1. Characteristics of Included Studies

In this review, 51 studies were included (Figure 1). Thirty-six (36/51, 70.6%) papers were original independent studies, and fifteen (15/51, 29.4%) papers were secondary analyses of prior surveys. Five (5/51, 9.8%) independent studies and ten (10/51, 19.6%) secondary studies included populations besides Chinese immigrants. Considering only Chinese immigrant participants, the sample size of independent studies ranged from 10 to 805 participants, and the sample size of secondary studies ranged from 120 to 2061. Six studies (6/51, 11.8%) were qualitative, two (2/51, 3/9%) were mixed-method, and forty-three (43/51, 84.3%) were quantitative. Thirty-three (33/51, 64.7%) were conducted in the United States of America, four (4/51, 7.8%) were conducted on participants in both Canada and America, and fourteen (14/51, 27.4%) were conducted in Canada (Table 1).

### 3.2. Dietary Characteristics

#### 3.2.1. Food Consumption

Eleven (11/51, 31.5%) studies reported that Chinese immigrants in Canada and the United States regularly consumed fruits or vegetables at an estimated amount of 2.7 to 3.6 servings/day. [5,26,29,30,32,42,44,47,49,53,61]. Studies reported a frequency of vegetable consumption ranging from at least weekly [53] to more than 3–5 times per week [42,49]. Chinese immigrants also self-reported a significant amount of vegetable intake [5,30]. Studies reported a frequency of fruit intake at least weekly [53] to more than 4–5 times weekly [42,49]. Five studies reported the quantity of combined fruit and vegetable consumption per day (3.5 servings per day, 3.6 servings per day, 566.3 g per day, 2.7 fruits or vegetables per day) [26,29,44,47,61]. One study reported fruit and vegetable consumption to be 2.7 and 2.6 servings, respectively, where each serving was equal to half a cup. The estimated consumption of fruits and vegetables among Chinese immigrants thus ranged from 2.7 to 3.6 cups per day [26,32,44,47,61].

Four (4/51, 7.8%) studies reported that grains are an important source of energy for Chinese immigrants in Canada and the United States [5,32,49,53]. Studies reported more than 90% of Chinese immigrants consumed grains more than 5 times a week [49] and, specifically, rice was consumed daily [5,53]. Grains were the main source of carbohydrates for recent Chinese immigrants [32].

Five studies (5/51, 9.8%) reported a low frequency or quantity of dairy consumption in Chinese immigrants in Canada and the United States, ranging from less than one serving per day to 1.6 servings/day [20,31,32,49,53,56]. Two studies found that Chinese immigrants consumed less than 1 serving of dairy a day, one found consumption of 1.6 cups per day, and yet another found that consumption of most dairy products occurred on a monthly basis [20,31,32,53]. One study reported that about 40% of immigrants consumed milk and alternatives less than or equal to 5 times per week [49].

Three (3/51, 5.9%) studies reported that most Chinese immigrants in Canada and the United States regularly consume meat and alternatives, including soybeans, fish, and eggs [32,49,53,56]. More than 75% of Chinese immigrants were reported to consume soybeans over 3 times per week [49]. Consumption of eggs, soy, and fish were reported to be higher in comparison with Caucasians [32,56], and consumption of meat and alternatives in general was reported to occur on a weekly basis [53].

#### 3.2.2. Macronutrient Intake

Five (5/51, 9.8%) studies reported varying daily carbohydrate consumption and inadequate fibre consumption among Chinese immigrants in Canada and the United States [20,32,34,54,62]. Three studies reported carbohydrates making up over 50% of total energy intake, [62] with daily intake ranging from 185 to 258 g per day [34,54,62]. Three studies reported fibre consumption of 12–14 g per day [20,34,54]. Four (4/51, 7.8%) studies reported quantities of protein consumption ranging from 1.1–1.5 g/kg/day, with one study reporting 89 g per day [37,54,62,63]. Chinese immigrants were found to consume more protein than Caucasians, South Asians, and Europeans [37,54]. Meat and fish were the largest source of protein in Chinese immigrants [63]. Seven (7/51, 13.7%) studies reported that fats were consumed regularly by Chinese immigrants in Canada and the United States, that the major source of fat intake was through cooking oils, and that there was a tendency to reduce fat among this population [42,49,54,57,60,61,62]. The daily quantity of fat consumption was reported to be 67 g per day in the general adult populations, and 54 to 57 g per day in males 60+ years of age, and 29 to 42 g per day in females 60+ years of age [54,62]. Studies reported daily consumption of vegetable oil and consumption of butter, margarine, or lard to be more than three times a week [49,53]. Thirty per cent of daily diet intake was reported to be derived from fats [62], with cooking oil being a major source [60,63], although Chinese immigrants had a tendency to reduce their intake of fat through methods such as limiting fried foods [42,57].

#### 3.2.3. Micronutrient and Caloric Intake

Four (4/51, 7.8%) studies reported a wide range of calcium intake in Chinese immigrants in Canada and the United States, ranging from less than 333 mg/day to 612 mg/day [20,35,48,50]. Three of these studies focused on Chinese women exclusively [35,48,50]. Ten (10/51, 19.6%) studies reported that Chinese immigrants in Canada and the United States had a lower daily caloric intake than Caucasians [8,19,21,28,32,34,54,61,63,65]. Caloric consumption was reported by two studies at 1592 kcal/day and 1736 kcal/day [8,34]. The dietary quality of Chinese immigrants was found to be 66.2/110 on the AHEI score, a scale for which values above 80 indicates a good diet and lower than 50 indicates a poor diet [19].

#### 3.2.4. Dietary Changes since Immigration

Eight (8/51, 15.7%) studies reported an increase in total consumption of food across all food groups and an adoption of Western food items in Chinese immigrants in Canada and the United States [5,20,23,36,46,52,59,64]. Three studies reported an increase in overall consumption of all food groups [23,52,59]. Two studies reported an increase in fruit and vegetable consumption specifically [36,64]. Four studies reported an increase in consumption of meat or dairy specifically [23,36,59,64]. Three studies reported that most Chinese immigrants consumed a traditional diet daily [5,23,46]. However, the frequency of consumption of traditional food is conflicting, with two studies reporting decreased consumption of traditional foods such as rice [52,64], and one reporting increased consumption of traditional foods [20].

Five (5/51, 9.8%) studies examined other dietary changes, such as adopting of food items (e.g., bread rolls, cakes, or pies), snacking between meals, drinking milk, and eating at fast-food restaurants [7,42,46,52,59,64]. Western dietary acculturation was found to be associated with higher-fat dietary behaviour [58,59]. Three studies reported that Chinese immigrants adopted certain Western foods, such as pizza, cereal, bread, pasta, or pies [23,52,59]. Three studies reported snacking between meals [42,59,64]. Three studies reported that breakfast was usually the first meal to be qualitatively altered (i.e., consuming peanut butter on toast) [42,46,59]. Additionally, studies found increased self-reported fast food and “junk food” consumption, the latter presumably referring to high-fat, high-calorie foods with little nutritional value according to a definition provided by the World Health Organisation [7,46,66]. However, one study found an increase in dietary variety and a decrease in high-fat foods since immigration [36].

### 3.3. Factors Influencing Dietary Behaviours

#### 3.3.1. Acculturation and Its Associations with Diet

Acculturation was reported to be associated with increased total energy intake, carbohydrate intake, and meat intake, although there were conflicting findings with regards to the association between acculturation and overall dietary quality, fruit and vegetable consumption, and fat consumption. Studies reported that an increased degree of acculturation or number of American friends was associated with increased total caloric intake [27,45], increased meat intake frequency, and increased beef, pork, and dairy intake [27,53]. There was no association reported between acculturation and consumption of fish or tofu [27]. Studies reported increased total carbohydrate and sugar intake with increased degree of acculturation [27,45]. The association between dietary quality and acculturation was conflicting. Various studies found acculturation to be associated with no significant effect on diet quality [23], that dietary quality was highest in less-acculturated participants [22], and that more-acculturated subjects had better dietary variety and adequacy [39,45,53]. Conflicting associations were also reported between acculturation and fruit and vegetable consumption [27,30,53,58]. Two studies reported increased fruit and vegetable consumption with acculturation [53,58]. However, one reported decreased expenditure on fruits and vegetables with acculturation, and yet another reported no change associated with acculturation [27,30]. Conflicting findings were reported on the association between acculturation and fat consumption [33,42,53,58,60]. Four studies found an association between acculturation and higher fat dietary behaviour [27,53,58,59]. One study reported an association of acculturation with decreased fat behaviour, and two others reported an association of acculturation with more fat-reducing behaviours and food items [42,45,58].

Five (5/51, 9.8%) studies reported conflicting findings on the effect of length of stay on dietary pattern changes of Chinese immigrants in Canada and the United States [24,28,31,47,53]. Three studies found that length of stay was not associated with changes in dietary patterns [28,31,53]. However, studies also reported that recent immigrants have a higher intake of meat and are more likely to use calcium supplements [31], that intake of fruit and vegetables decreased with length of stay [47], and that associations of a healthy traditional Chinese diet weakened with length of stay [24]. Studies also reported decreased added sugar consumption with length of stay, although there was no association between total sugar consumption and length of stay [20,34].

#### 3.3.2. Individual Factors

Twelve (12/51, 23.5%) studies reported that the dietary behaviours of Chinese immigrants in Canada and the United States were associated with individual demographics, individual preferences, and conceptualisations of health and nutritional awareness [7,16,23,24,25,33,38,41,42,53,59,60]. Individual demographics such as younger age, higher education, employment, and female gender were found to be associated with dietary acculturation [24,53,59]. Higher education and employment were associated with greater intake of energy and sugar [33]. Individual preferences and values influenced dietary behaviours, such as the desire for continued consumption of traditional foods such as ethno-cultural vegetables and the decision to decrease fat intake [23,42,55,60]. Personal awareness of nutritional information was associated with the adoption of Western dietary practices, and lack of information on healthful diets was a challenge for disease management [25,55]. The safety, quality, and freshness of products were major deciding factors in dietary behaviours and change [7,23,30,53,60]. Regularity, moderation, and the concept of maintaining yin/yang or “hot/cold” balance affected dietary behaviours as they were also seen as important aspects of healthy eating [7,41,42,51,60].

Six (6/51, 11.8%) studies reported associations between one’s personal health and life circumstances, such as nature of life experiences and language fluency with North American Chinese immigrants’ dietary behaviours [7,16,23,30,38,55]. One’s personal health condition also affects dietary behaviours, with conditions like pregnancy or having a disease associated with increased dairy consumption and healthier dietary behaviours, respectively [7,16]. Additionally, positive life events or experiences were associated with greater energy intake and healthier dietary behaviours [16,38]. Stress was associated with lower overall dietary intake, but a greater energy density and percentage of energy from fat [41]. The amount of time one has to prepare traditional and healthy meals is also an influencing factor [7,16,55]. Language was another influencing factor, acting as a barrier to acquiring safe food and underlying the decision to purchase cultural vegetables [23,30].

#### 3.3.3. Familial Factors

Eight (8/51, 15.7%) studies investigated the association of familial factors on dietary behaviours and changes in Chinese immigrants in Canada and the United States, including familial preferences and values, having young children in family, having older relatives and male partners, and household food environment [7,16,23,39,53,55,59,60]. Familial preferences and values have a major influence as individuals may seek family support in dietary decisions, and family preferences can be a facilitator or barrier to healthy eating [7,16,23,55,60]. One study found that older relatives and male partners prefer traditional foods, and two studies found that having young children was associated with the adoption of Western dietary practices [25,53]. Having young children was also associated with increased fruit and vegetable consumption [53]. Respondents in households with more high-fat foods exhibited higher fat-related dietary behaviour [58].

#### 3.3.4. Community Factors

Sixteen (16/51, 31.4%) studies investigated the association of community factors with the dietary behaviours of Chinese immigrants in Canada and the United States, including accessibility, cultural conceptualisations of health and eating, and community programs and people [5,7,16,23,30,33,40,41,42,43,51,52,53,55,56,57]. Accessibility of traditional foods is influenced by cost, availability, store location, and convenience of access. Chinese immigrants may be influenced in their decision to maintain traditional Chinese meals or eat healthily depending on the convenience of preparing traditional, healthy meals [7,16,39,52,53,55]. Limited access to traditional or healthy foods was reported to be associated with dietary acculturation or unhealthy diet behaviours [7,16,40,52,55]. Cost and socio-economic environment affects both the healthiness and acculturative degree of diets [7,23,53,55,60]. The cultural conceptualisations of health and diet also play an important role in dietary changes and behaviours. Eight studies reported that Chinese immigrants believe that diet is a very important part of health [5,7,30,40,41,42,51,56]. Specifically, the traditional concept of balancing “hot” vs. “cold” foods, or yin vs. yang foods, influences dietary decisions [7,24,42,51,60]. Finally, community nutrition education workshops and materials were identified as facilitators of healthy eating [16]. Neighbourhoods with higher immigrant populations were associated with healthier consumption and lower high-fat and processed food consumption [43].

## 4. Discussions

### 4.1. Summary of Findings

Chinese immigrants regularly consumed fruits or vegetables at an estimated amount of 2.7 to 3.6 servings/day. The consumption of milk and alternatives in Chinese immigrants was reported with values ranging from less than one serving per day to 1.6 servings/day. Grains, particularly rice, were an important source of energy for Chinese immigrants. Fiber consumption was inadequate according to national recommendations. Protein and fat consumption was sufficient. With immigration, Chinese immigrants showed an increased total consumption of food across all food groups and adoption of Western food items. Chinese immigrants also exhibited more snacking between meals and Westernisation of breakfast since immigration, although healthfulness of changes since immigration was variable. Total caloric intake, meat and alternatives intake, and carbohydrate intake increased with acculturation, although there were conflictive results regarding fruit and vegetable consumption and fat consumption. Individual factors associated with dietary behaviours of Chinese immigrants were individual demographics, preferences, and conceptualisations of health and nutritional awareness. Familial factors included familial preferences and values, having young children in family, and household food environment. Community factors included accessibility, cultural conceptualisations of health and eating, community programs and people.

### 4.2. Dietary Characteristics

The dietary behaviours of the Chinese immigrant population are discussed by comparing them against national dietary guidelines, while healthy eating is defined as dietary behavious that align with the United States or Canadian dietary guidelines. The findings of this review indicated that Chinese immigrants regularly consumed fruits and vegetables at an estimated amount of 2.7 to 3.6 servings/day. This amount is lower than the United States Department of Agriculture (USDA)’s Dietary Guidelines 2020–2025, recommending two cups of fruits and three cups vegetables for someone consuming 2000 calories per day [67]. Although these guidelines distinguish fruits and vegetables as different food groups, most findings from the included studies in this review do not. One study did find fruit and vegetable consumption to be 2.7 and 2.6 servings, respectively, where each serving was equal to half a cup—rendering each serving half of USDA standards. From this study and the comparatively low total consumption from other studies, it can be reasonably hypothesised that requirements from neither of the fruits or vegetables are met. These findings are consistent with a previous systematic review, which indicated only 0.5% to 20% of Asian American adults to be reaching the recommended fruit and vegetable consumption threshold [68]. Collectively, Canadians have been reporting a decreasing intake of fruits and vegetables, from 5.3 daily servings in 2004 to 4.5 daily servings in 2015, with a majority of people not meeting the guidelines in the 2007 Canada’s food guide [69]. Fruit and vegetable intake among both immigrants and the collective North American population is therefore in need of improvement.

Additionally, studies in this review suggested that grains are an important source of energy for Chinese immigrants. Rice is especially regarded as a culturally significant food, as evidenced by its daily consumption and cultural associations [5,41,53]. Rice is regarded by Chinese immigrants as a symbolically comforting food with nuanced cultural and historical meanings [41]. However, the cultural importance of rice consumption has been identified to be a barrier in diabetes management, as higher consumption of white rice has been associated with increased risk of developing type 2 diabetes [41,70]. More research is necessary to quantify daily intake of such foods and their purported health effects and to better inform healthcare providers and the public’s dietary decisions considering their cultural significance.

This review also found that consumption of milk and alternatives in Chinese immigrants was low, ranging from less than one serving per day to 1.6 servings/day. Chinese people in China consumed 11.8 g/day of milk, which also remains well below one serving [71]. By contrast, the consumption of milk and alternatives among Canadians is approximately 1.7 servings per day [72]. The Canadian food guide has moved away from strict serving guidelines for milk and alternatives, and instead encourages healthy proteins including low-fat dairy products [73]. These data would suggest increased efforts in choosing healthy dairy products and more attention to the source of protein rather than simply increasing intake of dairy. In addition, more research is needed on the adequacy of calcium consumption in Chinese immigrants, especially aging women prone to osteoporosis.

Findings from this review indicated that Chinese immigrants regularly consume meat and alternatives including soybeans, fish, and eggs, with higher consumption of the latter three compared to Caucasians. Although there is limited data on consumption of this food group among Chinese immigrants, the report of higher consumption of soybeans, fish, and eggs is in line with Canada’s food guide’s recommendations for protein sources, which also encourages plant-based proteins and lean meats. More data is needed to characterise meat and alternative consumption in Chinese immigrants, although the cultural preference for protein sources such as fish, eggs, and tofu presents itself as a facilitator of healthy dietary choices, as supported by Canada’s food guide.

Carbohydrate intake for Chinese immigrants was reported by one study in this review to be 50% of daily calorie intake, which is within the American dietary guidelines of 45–65% of daily intake [67]. Fiber consumption was reported by several studies in this review, with values ranging from 12–14 g per day. This is less than the minimal recommended 25 g/day for women and 38 g/day for men intake, according to Canadian health guidelines [67,74,75]. This insufficient intake of fibre is consistent with previous research on Chinese adults in China, where daily fibre intake was 9.7 g/day, as well as Canadian statistics, with most Canadians only meeting half the recommended amount [74,76]. Specifically, for Chinese immigrants, fibre intake can be encouraged by promoting whole grain noodles and brown rice instead of white rice. Encouragement of fruits and vegetables, along with the aforementioned focus on ethno-cultural vegetables, can aid fibre intake.

Sufficient protein and fat consumption by Chinese immigrants against USDA dietary guidelines was demonstrated by studies in this review [62]. However, focus should be placed on plant-based proteins and minimising saturated fats. Encouragement of culturally relevant foods such as tofu, and certain vegetables, as well as encouragement of fat-reducing behaviours can help in choosing good sources of fats and proteins for Chinese immigrants.

Included studies from this review reported an increase in consumption of various food groups, adoption of Western food items, as well as maintenance of traditional diet components after immigration. These changes have both the potential for healthy and unhealthy impact on dietary behaviours. The increased intake of fruits and vegetables for example, would aid in the consumption of good sources of fibre, protein, and vitamins and minerals, especially considering the inadequate fibre consumption found in this population by this review. However, the increase in certain dietary behaviours such as an increased intake of refined carbohydrates, high-fat, and fast food is unsupported by the Canadian and USDA food guides, and thus is unhealthy [67,73]. Previous research on the healthy immigrant effect would suggest that immigrants, although healthier than native-born Canadians upon immigration, experience a decrease in health status that is partly attributable to dietary changes [10].

### 4.3. Influencing Factors

Findings from this review indicated that acculturation was associated with increased total energy intake, carbohydrate intake, and meat intake. These associations may contribute to concepts identified by previous research, where immigrants experience a decrease in health status that is partly attributable to dietary changes after immigration [10]. This would indicate a need to educate Chinese immigrants on distinguishing healthy versus unhealthy sources of carbohydrates and proteins, as their consumption is likely to increase with acculturation. There were conflicting findings with regards to the association between acculturation and overall dietary quality, fruit and vegetable consumption, and fat consumption. This may be in part due to inconsistencies in the measurement of acculturation. Studies used different indicators, such as number of American friends, media consumption, or English proficiency. The two-dimensional acculturation model where participants are non-exclusively assessed according to their maintenance of traditional culture and their adaptation to Western culture should be considered for future studies [77].

This review also indicated that individual factors influencing dietary behaviours of Chinese immigrants were associated with lack of time and nature of migration life experiences. The lack of time was identified as a barrier in other immigrant groups as well, such as Chinese immigrants in Spain as well as Arab Muslim immigrants to Canada [78,79]. The lack of time to prepare traditional meals leads to increased consumption of more easily prepared and accessible Western meals. Health promotion is necessary to facilitate healthy dietary decisions under time constraints, whether that involves choosing healthier Western alternatives, rather than refined carbohydrates and high-fat foods, or enabling easier access to healthy traditional foods. Interestingly, positive experiences also influenced healthier dietary behaviours, whereas stress was associated with a greater energy density and percentage of energy from fat. Stress related to immigration and acculturation may also be a contributing factor. This is consistent with previous research highlighting lack of social relationships, busier lifestyle, and higher stress levels as contributors to unhealthy dietary changes in immigrant women [80]. This highlights the importance of addressing stress and mental health among Chinese immigrants, as they are important determinants of dietary practices, which can in turn influence chronic health outcomes [80].

This review indicated that having young children in family influenced individual and familial dietary behaviors. Children’s preferences have a particular influence on immigrant food choices, as it was identified as a factor in dietary choices of Chinese immigrants in Spain [79]. Children’s preferences were also identified as an influencing factor in a study of Arab Muslim immigrant mothers in Canada [78]. Educational interventions on healthy dietary choices targeted at family members, specifically children, could encourage healthy eating behaviours in Chinese immigrants in North America.

Finally, this review indicated that cultural conceptualisations of health and eating as well as neighbourhood makeup were important community factors influencing Chinese immigrants’ dietary behaviors. Cultural concepts of yin and yang, as well as hot and cold foods were important influences. Traditional Chinese Medicine dictates that certain foods are hot or cold by nature, and their balance is crucial in maintaining health [42]. Health promotion methods that account for both traditional Chinese conceptualisations of a healthy diet along with standard Western evidence-based models are necessary to promote healthy dietary behaviours among Chinese immigrants. Another interesting finding was that neighbourhoods with higher immigrant populations were associated with healthier consumption and lower high-fat and processed food consumption among Chinese immigrants. This finding is consistent among native-born Canadians, who showed healthier behaviours and better health outcomes in areas with higher immigrant density [81]. This indicates potential for immigrant food practices to be beneficial for the larger population as well [78,79].

### 4.4. Implications

There are several implications of our findings, especially on how they can be used to influence healthy eating, which should be aligned with current Canada Food Guide and USDA recommendations. Firstly, efforts can be made among Chinese immigrant individuals to increase their fruit and vegetable intake and fibre intake, resist unhealthy acculturative changes (i.e., increased consumption of high-fat foods), and aid family members to do the same. Secondly, considering many Chinese immigrants seek nutritional information from family physicians [42], health care providers can enable healthy dietary changes using culturally sensitive methods. Diet plans and recommendations can include ethno-cultural vegetables and integrate traditional concepts where appropriate [82]. The sociocultural importance of certain staples and their contribution to well-being may also be recognised so that providers can work with individuals to find appropriate replacements or compromises. Healthcare providers can also take care to acknowledge and address the role of mental well-being in dietary decisions, with special consideration of the unique stressors immigrant groups may face. Thirdly, given their importance to this population, Chinese immigrant community organisations, especially those related to health, can play an important role in dietary education [83]. These can include cooking workshops or nutrition information sessions focused on encouraging fruits and vegetables, fibre, and avoiding saturated fats, among other healthy dietary behaviours. Finally, given that children are key influencers in the home diet, schools can create environments that support healthy eating and encourage children to bring home new health knowledge. Decisions regarding policies on immigration and neighbourhood planning can also consider the health benefits associated with areas of high immigrant density, improve access to ethnic foods, and promote sharing of cultural food practices. In conclusion, as has been proposed by previous studies, individuals, healthcare providers, and organisations can through their collective efforts, improve health outcomes for the population [84]. These findings and implications have the potential to be applicable to Chinese immigrants in other high-income Western countries, although the cultural, social, economic, and dietary differences between the countries should be considered prior to any generalisation of findings.

### 4.5. Limitations of This Review and Recommendations for Future Studies

There are some limitations in this review. Firstly, a variety of methods for measuring dietary intake are used across studies. Food groups and macronutrients were measured as servings, percentages, grams, and frequencies. The definition of a serving also varied, making it difficult to compare study findings. These discrepancies were further complicated by the fact that many studies grouped foods together (i.e., fruits and vegetables as one), while other studies did not. Furthermore, the Canada Food Guide and the USDA have different guidelines and food group categories, with the former providing no strict serving recommendations and no distinction between fruit and vegetable recommendations. Additionally, different food frequency questionnaires with different numbers of items and lengths of recall were used. Such methods rely heavily on participant memory, which is subjective and liable to error. Future studies should further develop the validity and reliability of food frequency questionnaires and other measurement tools to improve the rigor of dietary research. Secondly, there is a lack of studies undertaken on sodium, potassium, and calcium intake in Chinese immigrants. Given the importance of sodium and potassium levels in blood pressure regulation, and evidence of hypertension as the strongest risk factor for stroke in Chinese people, this aspect of dietary intake requires more investigation [85,86]. Dietary studies of calcium also need to be repeated in larger populations to discern whether the lower consumption of milk and alternatives among Chinese immigrants is a cause for concern. Finally, there is a lack of longitudinal cohort studies, case-control studies, and intervention studies on the nutrition of Chinese immigrants. The majority of data comes from cross-sectional studies relying heavily on self-reporting of dietary consumption over a few days. In the future, interventions studies, especially randomised controlled trials, should be conducted to investigate the role of promising dietary behaviours or specific dietary interventions on health outcomes in Chinese immigrants to North America.

## 5. Conclusions

This review characterises dietary behaviours and influencing factors of the Chinese immigrant population in Canada and the United States. This review found that fruit and vegetable, fibre, and dairy consumption among Chinese immigrants was generally insufficient. Although protein, fat, and carbohydrate intake was generally sufficient, efforts can be taken to ensure healthy sources are selected. Dietary acculturation was also observed in the Chinese immigrant population in this review, and although not an inherently negative change, efforts can be undertaken to ensure that healthy Western foods are adopted. Dietary behaviours in Chinese immigrants are influenced by everything from traditional health beliefs, time, and accessibility, to family members and neighbourhoods. It is important for healthcare providers and nutritionists to remain culturally sensitive when providing dietary recommendations. This can be achieved through encouragement of healthy ethnocultural foods and incorporating traditional health beliefs into Western evidence-based principles.

## Figures and Tables

**Figure 1 nutrients-14-02166-f001:**
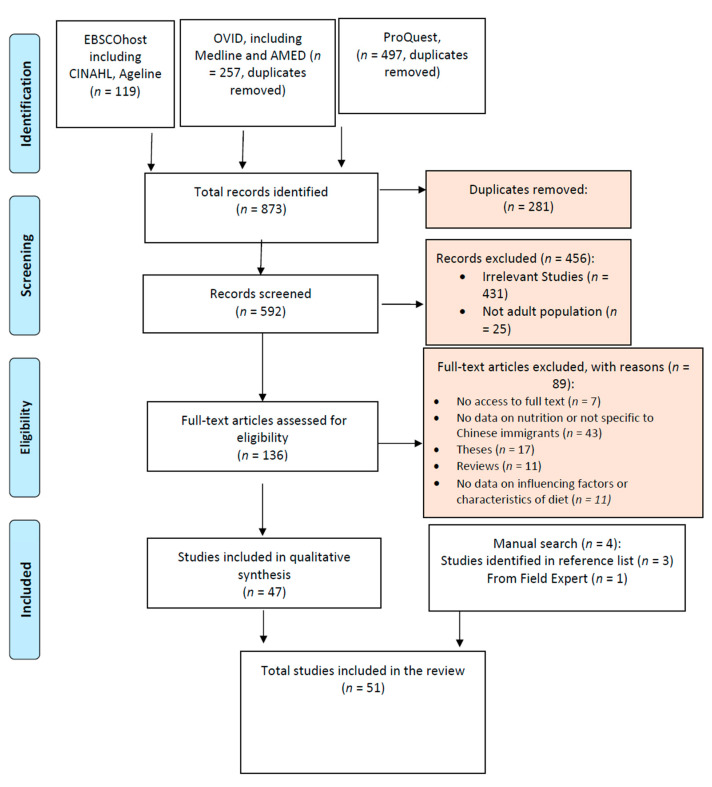
Study selection flow diagram.

**Table 1 nutrients-14-02166-t001:** Characteristics of included studies.

	Author Year	Research Setting	Research Design	Sampling
1	Rodriguez et al. (2020) [19]	USA, community	Quantitative, cross-sectional secondary analysis from the Mediators of Atherosclerosis in South Asians Living in America (MASALA) and the Multi-ethnic study of atherosclerosis (MESA), which were longitudinal studies	3927 + 889 participants free of CVD
2	Beasley et al. (2019) [8]	New York City, USA	Quantitative, cross-sectional survey	1973 Chinese immigrants, as part of Chinese American Cardiovascular Health Assessment
3	Chai et al. (2019) [20]	Delaware, USA, university setting	Quantitative cross-sectional survey	172 Asian students, students, 108 from China or Taiwan
4	Kirshner et al. (2019) [21]	New York City, USA, community	Quantitative, cross-sectional survey	2071 Chinese American New Yorkers, data from the Chinese American Cardiovascular Health Assessment
5	Higginbottom et al. (2018) [7]	Alberta, CAN, community	Quantitative, ethnography and interviews	23 Chinese Canadian perinatal women
6	Zou (2018) [16]	Greater Toronto Area, CAN, telephone	Qualitative, telephone interviews	30 aged Chinese-Canadian participants who received the DASHNa-CC
7	An (2017) [22]	United States, online	Quantitative, cross-sectional survey	505 Chinese living in US, online
8	Liu et al. (2017) [23]	Greater Toronto Area, CAN, community	Mixed methods, qualitative focus group, and quantitative small-scale cross-sectional survey	12 female Chinese immigrants, who had been in Canada for 10 years or less
9	Lu et al. (2017) [24]	Vancouver, Toronto, Halifax and St Catherines, CAN, community	Quantitative, cross-sectional survey	100 Chinese immigrants aged 25+
10	Wang et al. (2017) [23]	California, USA	Quantitative, cross-sectional	2122 Asian adults, 658 of whom were Chinese
11	Zou (2017) [25]	Greater Toronto Area, CAN, classroom and community setting	Quantitative RCT, intervention received DASHNa-CC, control received usual care	61 Chinese Canadians 45+ with hypertension but not on medications
12	Yi et al. (2016) [26]	New York City, USA, community	Quantitative, cross-sectional data obtained from New York City Community Health Survey	555 Chinese America adults with hypertension, 144 South Asian adults with hypertension, 5987 Non Hispanic white adults with hypertension
13	Tseng et al. (2015) [27]	Philadelphia, USA, community	Quantitative, longitudinal study	312 Chinese immigrant women
14	Corlin et al. (2014) [28]	Boston, USA, community-based	Quantitative, cross-sectional	147 Chines immigrants and 167 US born whites participating in Community assessment of Freeway Exposure and Health study
15	Wyat et al. (2014) [29]	New York City, USA, community	Quantitative longitudinal, surveys over 4 years.	805 Chinese Americans aged 65+, foreign-born
16	Adekunle et al. (2013) [30]	Greater Toronto Area, CAN	Quantitative, cross-sectional survey, predictive factor analysis	250 Chinese Canadian respondents, representing household averaging four people
17	Wong et al. (2013) [31]	New York City, USA, community	Quantitative cross-sectional	125 older (50+) Chinese persons
18	Tam et al. (2012) [32]	Toronto and Vancouver, CAN, community	Quantitative, Cross sectional	1050 postmenopausal Canadian women, 421 of whom were recent Chinese migrants, 216 of whom migrated to the West before age 21
19	Tseng et al. (2012) [33]	Philadelphia, USA, community	Quantitative, Cross-sectional surveys	437 healthy premenopausal Chinese Immigrant women
20	Alonge et al. (2011) [34]	Houston, Texas, USA	Quantitative, Cross-sectional surveys	213 Chinese, Mexican and Nigerian immigrants, 52 of whom were Chinese.
21	Lv et al. (2011) [35]	USA, community	Quantitative, Quasi-experimental study with a nested design and pre- and post design	151 first generation Chinese American mothers between 35 and 55
22	Rosenmoller et al. (2011) [36]	CAN, community	Quantitative, cross-sectional sub-study of the Multi-cultural Community health assessment Trial, study cohort	120 Chinese-born people living in Canada
23	Tam et al. (2011) [37]	Toronto and Vancouver, CAN, community	Quantitative, cross-sectional	1051 postmenopausal Canadian women, 383 of whom were recent Chinese migrants, 156 of whom migrated to the west before age 21
24	Tseng et al. (2011) [38]	Philadelphia, USA, community	Quantitative cross-sectional surveys	436 healthy premenopausal Chinese Immigrant women
25	Liu et al. (2010) [39]	Philadelphia, USA	Quantitative, cross-sectional	243 Chinese Americans who were part of study on diet and breast density
26	Bell et al. (2009) [40]	British Columbia, CAN, professionally facilitated support group	Qualitative, ethnography	96 Chinese Canadian participants in cancer support groups
27	Chesla et al. (2009) [41]	USA	Qualitative, comparative interview	20 Chinese American couples, one with diabetes
28	Kwok et al. (2009) [42]	Toronto, CAN community	Quantitative, cross-sectional survey	106 Chinese Canadians
29	Osypuk et al. (2009) [43]	Four USA cities, community	Quantitative, secondary analysis from the Multi-ethnic study of atherosclerosis, which was a longitudinal study	1902 Study participants
30	Washington et al. (2009) [5]	California, USA, two Chinese senior care facilities	Qualitative semi-structured interviews	13 participants, aged 65 years or older, who had a diagnosis of type 2 diabetes
31	Hislop (2008) [44]	Vancouver, CAN	Quantitative, cross-sectional	504 Chinese adult immigrants
32	Kandula et al. (2008) [45]	USA: Baltimore, Chicago, Forsyth County, LA, NYC, St Paul	Cross-sectional data from Multi-Ethnic Study of Atherosclerosis	1255 Hispanics and 737 Chinese participants
33	Lu et al. (2008) [46]	Western Canada, community	Qualitative, semi-structured interviews	10 individuals
34	Taylor et al. (2007) [47]	Seattle, USA, community	Quantitative, cross-sectional survey	495 Chinese immigrants
35	Babbar et al. (2006) [48]	New York City, USA, family care center	Mixed Methods, concurrent triangulation of cross-sectional study and qualitative surveys	300 Chinese American Women
36	Fang et al. (2006) [49]	New York City, USA, hospital based	Quantitative, case control study	187 foreign-born Chinese stroke cases and 204 controls matched
37	Walker et al. (2006) [50]	USA, community	Quantitative, development of prognostic model	359 Chinese American women, ambulatory, ages 20–90
38	Liang et al. (2004) [51]	Washington, DC, USA, community	Qualitative, focus groups	54 Chinese American women aged 50+
39	Lv et al. (2004) [52]	Pennsylvania, USA, community	Quantitative, cross-sectional self-administered survey	399 Chinese Americans, 18+ in Pennsylvania
40	Lv et al. (2003) [53]	Pennsylvania, USA	Quantitative, cross-sectional survey	399 Chinese Americans, 18+ in Pennsylvania
41	Kelemen et al. (2003) [54]	Hamilton, CAN, community	Quantitative, development of a tool (involved multiple 24 h recalls, items tabulated, assessed, and included into existing tool)	74 immigrants, 25 of whom were Chinese
42	Satia-Abouta et al. (2002) [55]	Seattle, Vancouver, USA	Quantitative, Secondary analysis of data from Chinese Women’s Health Project, cross sectional	244 adult females of Chinese ethnicity
43	Wu et al. (2002) [56]	Los Angeles County, USA	Quantitative case-control study	523 cases, Asian American women between ages of 25–74 at time of diagnosis of breast cancer were identified through the LA Cancer Surveillance program. 160 were Chinese. 594 controls were selected from the neighbourhood. 228 were Chinese
44	Liou et al. (2001) [57]	New York City, USA, community	Quantitative, cross-sectional survey	600 health Chinese Americans between 25 and 70 years of age
45	Satia et al. (2001) [58]	Seattle, USA and Vancouver, CAN community	Quantitative, Secondary analysis of data from Chinese Women’s Health Project, cross-sectional	244 adult females of Chinese ethnicity
46	Satia et al. (2001) [59]	Seattle, USA, and Vancouver, CAN	Quantitative, Secondary analysis of data from Chinese Women’s Health Project, including cross-sectional survey, development of a measurement tool	244 adult females of Chinese ethnicity
47	Satia et al. (2000) [60]	Seattle, USA, community	Qualitative interviews and focus groups, qualitative groundwork to develop quantitative dietary survey tool	42 Chinese American women
48	Whittemore et al. (1995) [61]	USA: LA, San Francisco, Hawaii, CAN: Vancouver, Toronto In community and lab	Quantitative case control study	1655 prostate cancer cases were identified through cancer registries in Hawaii, LA, SF, Vancouver, and Ontario Cancer Registry, 283 of whom were Chinese Americans. 1645 controls, 272 of whom were Chinese Americans
49	Choi et al. (1990) [62]	Boston, USA	Quantitative, cross-sectional surveys	346 healthy elderly Chinese aged 60–96
50	Whittemore et al. (1990) [63]	USA: LA, San Francisco, Vancouver CHINA: Hangzhou, Ningbo Hospitals	Quantitative, case-control study	805 Chinese North American patients were identified from the British Columbia Cancer Registry
51	Newman et al. (1982) [64]	New York City, USA, community	Quantitative, cross-sectional surveys	102 Chinese immigrant mothers

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
