# Peer review of "Dietary Characteristics and Influencing Factors on Chinese Immigrants in Canada and the United States: A Scoping Review"

_nutrients, 2022, doi:10.3390/nu14102166_

Round 1

Reviewer 1 Report

Dear authors:

please

1-improve the quality of figure 1

2-may if make table 1 shorter and move the unnecessary data to the supplementary would be better for the readers

Reviewer 2 Report

Good work in an hot topic.

I suggest to add and comment in discussion this reference:

10.1002/pbc.25543

an hematological experience to utility assumptions from a clinician's perspective

Reviewer 3 Report

This paper reviewed the literature to determine dietary characteristics that influence the dietary choices of Chinese who immigrate to Canada and the United States. The title calls it a scoping review, while the body of the paper says it is a systematic review. These are not the same. Please see this article to explain the difference and correct the title or body of the paper.

Munn, Z., Peters, M.D.J., Stern, C. et al., Systematic review or scoping review? Guidance for authors when choosing between a systematic or scoping review approach. BMC Med Res Methodol 18, 143 (2018). https://doi.org/10.1186/s12874-018-0611-x

What PRISMA guidelines were used as these changed in 2020?

The authors conclude that healthcare providers must make sure Chinese immigrants choose healthy Western foods. Yet, the frequency with which healthcare providers see Chinese immigrants is probably infrequent, so the ability of this group to make an impact is minimal. The authors do not discuss how often Chinese immigrants see healthcare providers nor alternative ways to educate Chinese immigrants, which may have a more significant influence.

In the Discussion section under Summary of findings, the authors use the term grains (fourth line) as an important energy source. Are these whole grains or processed grains like white rice?

The authors use the term Westernization or Western diet but don’t provide a succinct definition. They often say the immigrant’s meals and food selections are “Westernized.” What does that mean? Do they become high in sodium? The Chinese diet is high in this nutrient. Does it become high in saturated fat or sugar?

The authors state the immigrant’s diet should be healthier but don’t define healthy. Many traditional Chinese foods are not “healthy.”  Zhang, R., Wang, Z., Fei, Y., Zhou, B., Zheng, S., Wang, L., Huang, L., Jiang, S., Liu, Z., Jiang, J., & Yu, Y. (2015). The Difference in Nutrient Intakes between Chinese and Mediterranean, Japanese and American Diets. Nutrients, 7(6), 4661–4688. https://doi.org/10.3390/nu7064661

In discussion section 4.2, each paragraph starts with “Findings of this review.”

The authors lump fruits and vegetables together, yet according to the USDA’s MyPlate, these are separate food groups, so they should be separated in the paper.

For 2000 calories, the USDA recommends 2 cups fruit and 3 cups vegetables. Again, this should be correctly stated in the paper.

Round 2

Reviewer 3 Report

Is it a scoping review or a systematic review? Not the same as pointed out in a reference I provided. The authors acknowledged this. The word systematic appears 8 times in the paper still; of these 3 appear in the references, one is the heading from the journal, and one is under section 2.3. The other three should be changed.

Again, the authors use the term “healthy eating” but never adequately define this term. This phrase needs to be defined since they make dietary recommendations for Chinese immigrants. Healthy eating would be meeting the US and Canadian Dietary Guidelines.

Abstract:  What are Western biomedical principles? Biomedical means relating to biology and medicine. What are unique Western biology and medicine principles? How does this apply to diet and this paper?

Section 4.2: What are the American National Guidelines? The American Dietary Guidelines and MyPlate distinguish fruits and vegetables as separate groups.

Section 4.2: What is junk food? There is no standard definition for this term, and it is commonly used by the general public and not in the scientific literature. The authors need to be more explicit in their definition, for example, energy-rich, nutrient-poor.

Implications:  The consumption of snacks is not necessarily unhealthy. It is the food that people select for a snack that is unhealthy. The immigrants should select nutrient-rich foods which tend to be less calorically dense if they are going to snack.

Implications: Not all processed foods are bad. Again, the authors try to overgeneralize when they need to be more specific. For example, soy sauce, a mainstay of the Chinese culture, is processed, but that is not the type of food the authors are referring to in this section.

Section 3.24: The authors define the Westernized diet using studies in the review. This diet includes characteristics such as (1) adopting Western food items such as bread rolls, cakes, or pies, (2) snacking between meals, (3) drinking milk, and (4) eating at fast-food restaurants (Higginbottom, Vallianatos, Shankar, Safipour, & Davey, 2018; Kwok et al., 2009; Lu, Sylvestre, Melnychuk, & Li, 2008; Lv & Cason, 2004; Newman, 1980; Satia, Patterson, Kristal, Hislop, & et al., 2001). Generally, the characteristics of the Western diet are high sugar, salt, and saturated fat. Fast-food restaurants were first introduced in the United States but are now available in China too, so it is no longer unique to the “Westernized” diet. Fast foods tend to be high in salt and saturated fat and poorer in nutrient density. The problem is the use of the term Westernized when the Chinese immigrants are becoming acculturated to the United States and Canada and the foods more commonly available in these countries.
